# The Emerging Role of Colchicine to Inhibit NOD-like Receptor Family, Pyrin Domain Containing 3 Inflammasome and Interleukin-1β Expression in In Vitro Models

**DOI:** 10.3390/biom15030367

**Published:** 2025-03-03

**Authors:** Tri Astiawati, Mohammad Saifur Rohman, Titin Wihastuti, Hidayat Sujuti, Agustina Endharti, Djanggan Sargowo, Delvac Oceandy, Bayu Lestari, Efta Triastuti, Ricardo Adrian Nugraha

**Affiliations:** 1Doctoral Program of Medical Science, Faculty of Medicine, Brawijaya University, Malang 65145, Indonesia; agustin_almi@ub.ac.id; 2Department of Cardiology, Doctor Iskak General Hospital, Tulungagung 62233, Indonesia; 3Department of Cardiology and Cardiovascular Medicine, Faculty of Medicine, Brawijaya University, Malang 65145, Indonesia; ippoenk@ub.ac.id; 4Cardiovascular Research Centre, Brawijaya University, Malang 65145, Indonesia; 5Department of Biomedical, Nursing Science, Faculty of Medicine, Brawijaya University, Malang 65145, Indonesia; titinwihastuti@gmail.com; 6Department of Biochemistry, Faculty of Medicine, Brawijaya University, Malang 65145, Indonesia; hidayatsujuti.fk@ub.ac.id; 7Department of Cardiology and Vascular Medicine, Faculty of Medicine, Brawijaya University, Malang 65145, Indonesia; djanggan@ub.ac.id; 8Division of Cardiovascular Science, The University of Manchester, Manchester Academic Health Science Centre, Manchester M13 9PT, UK; delvac.oceandy@manchester.ac.uk; 9Department of Pharmacology, The University of Manchester, Manchester Academic Health Science Centre, Manchester M13 9PT, UK; bayu.lestari@manchester.ac.uk; 10Department of Pharmacy, Faculty of Medicine, Brawijaya University, Malang 65145, Indonesia; efta.triastuti@ub.ac.id; 11Department of Cardiology and Vascular Medicine, Faculty of Medicine, Universitas Airlangga—Dr. Soetomo General Academic Hospital, Surabaya 60286, Indonesia; ricardo.adrian.nugraha-2019@fk.unair.ac.id

**Keywords:** ASC-NLRP3, inflammasome, colchicine, IL-1β, ischemia

## Abstract

While the beneficial effects of colchicine on inflammation and infarcted myocardium have been documented, its impact on cardiac fibroblast activation in the context of myocardial infarction (MI) remains unknown. This study aimed to investigate the effect of colchicine on the regulation of NOD-like receptor family, pyrin domain containing 3 (NLRP3) inflammasome activation and Interleukin-1β (IL-1β) expression in fibroblasts. 3T3 fibroblasts were exposed to 600 μM CoCl_2_ for 24 h to simulate hypoxia, with normoxic cells as controls. Colchicine (1 μM) was administered for 24 h. ASC-NLRP3 colocalization and IL-1β expression were evaluated using immunofluorescence and flow cytometry, respectively. Data were analyzed using t-tests and one-way ANOVA with post hoc tests. Hypoxia treatment significantly induced apoptosis-associated speck-like protein containing a CARD (ASC)-NLRP3 colocalization (*p* < 0.05). Colchicine treatment of hypoxic 3T3 cells reduced ASC-NLRP3 colocalization, although this reduction was not statistically significant. Additionally, IL-1β expression was significantly inhibited in colchicine-treated hypoxic 3T3 cells compared to those treated with placebo (*p* < 0.05). The findings of this study indicate that colchicine treatment inhibits the activation of the NLRP3 inflammasome by disrupting the colocalization of ASC and NLRP3, thereby reducing IL-1β expression in CoCl_2_-treated 3T3 cells.

## 1. Introduction

Epidemiological data demonstrated that 58% of mortality associated with cardiovascular diseases (CVDs) has occurred in Asia [1]. Myocardial infarction, commonly referred to as a heart attack, transpires when blood flow to the myocardium is obstructed, resulting in damage or necrosis of the cardiac muscle tissue [2]. Ischemic heart disease, encompassing acute myocardial infarction, represents a critical condition necessitating prompt medical intervention to mitigate further myocardial damage and to diminish the risk of complications such as heart failure and other potentially fatal outcomes [3]. The onset of heart failure is associated with elevated morbidity and mortality rates, thereby exacerbating the social and economic burden [4]. In addition to myocardial necrosis, ventricular remodeling is intimately linked to the pathogenesis of heart failure in patients following myocardial infarction [5].

Inflammation, a multifaceted biological process, is integral to the pathogenesis of heart disease, including cardiac remodeling following MI [6,7]. Under ischemic conditions/hypoxia, reduced blood flow to the myocardium can cause significant tissue injury and initiate an inflammatory response [8]. While this inflammatory response is essential for cardiac repair following acute myocardial infarction, excessive inflammation can result in adverse ventricular remodeling [9,10]. This inflammatory activity constitutes a critical aspect of the pathophysiological mechanism in cardiac ischemia, involving various regulatory elements, including interleukin-1 beta (IL-1β) [11] and the NOD-like receptor family pyrin domain-containing 3 (NLRP3) inflammasome complex [12], which serves as a principal mediator of sterile inflammation post-acute myocardial infarction [13]. The NLRP3 inflammasome is crucial in the innate immune system, which mediates caspase-1 activation and the secretion of proinflammatory cytokines IL-1β/IL-18 in response to microbial infection and cellular damage [14]. This process culminates in the release of pro-inflammatory mediators that evoke pro-inflammatory responses and attract inflammatory cells to the infarcted area, contributing to ventricular remodeling [15]. The activation of the NLRP3 inflammasome leads to the production of pro-inflammatory cytokines, notably interleukin-1β (IL-1β), which contribute to tissue damage and adverse cardiac remodeling post-MI.

Colchicine, a natural alkaloid traditionally employed in the management of gout [16], has recently gained recognition as a potential therapeutic agent for ischemic heart disease [17]. As an anti-inflammatory compound, colchicine can inhibit the production of IL-1β and the activation of the NLRP3 inflammasome [18], thereby attracting considerable interest within the scientific community. In the context of acute myocardial infarction (AMI), colchicine administration has demonstrated efficacy in reducing infarct size, enhancing cardiac function, and modulating inflammatory responses [16]. Notably, the Colchicine Cardiovascular Outcomes Trial (COLCOT) reported a 23% reduction in the risk of adverse cardiovascular events among patients with acute coronary syndrome treated with colchicine, compared to those receiving a placebo [19]. While clinical studies have shown promising results in reducing cardiovascular events in patients with coronary artery disease, the precise mechanisms by which colchicine exerts its effects on cardiac inflammation, particularly in the context of ischemia, remain to be fully elucidated.

The speck itself is not the NLRP3 inflammasome but is instead a dynamic structure that may amplify the NLRP3 response to weak stimuli by facilitating the formation and release of small NLRP3:ASC complexes, which in turn activate caspase-1 [20].

This study aims to investigate the effects of colchicine on NLRP3 inflammasome activation and IL-1β expression in a cellular model of hypoxia using 3T3 fibroblasts. We hypothesize that colchicine may attenuate inflammasome activation by interfering with the assembly of its components, specifically the interaction between ASC and NLRP3 proteins.

## 2. Materials and Methods

### 2.1. Ethical Statement

This study was approved by the Health Research Ethics Committee of the Faculty of Medicine, Brawijaya University, Indonesia (Certificate No. 182/EC/KEPK-S3/07/2023) under the name of Tri Astiawati as principal investigator. Trial registration: https://clinicaltrials.gov/study/NCT06426537. Unique identifier: NCT06426537. Study started on 20 October 2022 and completed on 20 November 2023.

### 2.2. Study Design

The 3T3 fibroblast cells were animal-derived and randomly allocated to different treatment groups, including control, hypoxia, colchicine, and combined colchicine–hypoxia groups. Randomization was performed using a random number generator to ensure the unbiased allocation of cells into each experimental condition.

Cells were randomly allocated to four experimental groups: control (non-hypoxic cells), colchicine (non-hypoxic cells + colchicine for 24 h), CoCl_2_ (hypoxic cells), and CoCl_2_ + colchicine (hypoxic cells + colchicine for 24 h) [21].

To minimize bias during data analysis, the samples were labeled with coded identifiers that were unknown to the investigators until the analysis was completed. This procedure was maintained throughout the experiment to ensure the objective evaluation of the results.

### 2.3. Cell Culture and Treatment

The 3T3 fibroblast cells used in this study were obtained from the European Collection of Authenticated Cell Cultures (ECACC, catalog no. 306-05A, RRID:CVCL_0594, Porton Down, Salisbury, Wiltshire, UK). The cells were authenticated using short tandem repeat (STR) profiling to confirm their identity, and they were regularly tested for mycoplasma contamination to ensure the integrity of the experimental results. The 3T3 cells were cultured in 6-well plates (80,000 cells/well) in Dulbecco’s Modified Eagle Medium (DMEM, Gibco, catalog no. 11995065, Thermo Fisher, Waltham, MA, USA) supplemented with 10% fetal bovine serum (FBS, Gibco, catalog no. 16000044) and 1% penicillin-streptomycin (Gibco, catalog no. 15140122, Thermo Fisher, Waltham, MA, USA). Hypoxia was induced using 600 μM CoCl_2_ for 24 h, while colchicine was administered at 1 μM for 24 h [22].

Hypoxia induction and colchicine treatment hypoxic conditions were simulated using 600 μM Cobalt(II) Chloride Hexahydrate (CoCl_2_, Sigma-Aldrich, St. Louis, MO, USA, catalog no. C8661-25G, CAS: 7791-13-1) for 24 h. The optimal concentration of 600 μM CoCl_2_ was established based on prior studies [23]. Colchicine (Sigma-Aldrich, catalog no. C9754, CAS: 64-86-8) was administered at a final concentration of 1 μM in DMEM supplemented with 10% FBS and 1% penicillin–streptomycin. The media were also supplemented with 1% PEST (Sigma-Aldrich, St. Louis, MO, USA). All the incubations were performed at 37 °C with 5% CO_2_, incubated for 24 h.

### 2.4. Co-Localization

For the co-localization of ASC/NLRP3 inflammosome, the cells were seeded at 2 × 10^4^ cells per well on 8-well glass slides (Sarstedt, Hildesheim, Germany) in 400 µL media and incubated for 20 h at 37 °C with 5% CO_2_. Thereafter, the old media were removed, and new media were added. The cells were further incubated for 48 h at 37 °C with 5% CO_2_. The cells were fixed with 4% PFA, permeabilized for 10 min with 0.4% Triton, blocked for 1 h with PBS with 2% BSA (Bio-Rad Laboratories Inc., Hercules, CA, USA), and incubated with the primary antibodies, anti-ASC (sc-514414, Santa Cruz Biotechnology, Inc., Dallas, TX, USA) and anti-NLRP3 (ABF23, MilliporeSigma, Burlington, MA, USA), diluted 1:100 in PBS with 0.1% BSA at 4 °C overnight. We used Alexa FluorTM 488 goat anti-rabbit IgG (H + L) (A11034, Invitrogen, Waltham, MA, USA) as secondy antibody, which were added at concentrations of 1:2000.

### 2.5. Statistical Analysis

A priori power analysis was conducted using G*Power 3.1 software to determine the sample size. With an effect size of 0.8, an α error probability of 0.05, and a power (1-β error probability) of 0.8. We used 5 replicates per group to account for potential experimental loss. A power analysis was conducted to determine the appropriate number of replicates needed to detect significant differences between the treatment groups. The analysis suggested that a minimum of 5 replicates per group would be required to achieve a statistical power of 80% at a significance level of *p* < 0.05. This sample size was chosen to ensure that the experimental results were robust and reliable.

## 3. Results

### 3.1. NLRP3 Inflammasome

To evaluate the impact of colchicine treatment on the activation of the inflammasome pathway, a co-localization study of ASC and NLRP3 in 3T3 cells was conducted using the immunofluorescence method. Our data revealed that cobalt chloride treatment significantly induced co-localization of ASC and NLRP3 in 3T3 cells, as indicated by the presence of yellow dots or clusters within the cytoplasm (*p* < 0.05). Additionally, colchicine administration resulted in a reduction in the percentage of 3T3 cells exhibiting positive ASC-NLRP3 co-localization, although this reduction was not statistically significant. Representative images of each group are presented in Figure 1, while the quantification of cells with positive ASC-NLRP3 co-localization is illustrated in Figure 2.

### 3.2. IL-1β Expression

Our data showed that CoCl_2_ treatment significantly elevated the expression of IL-1β compared to normoxic 3T3 cells (62.58 ± 1.41% vs. 52.02 ± 7.89%, *p* < 0.05). Colchicine treatment significantly reduced IL-1β expression in hypoxic cells compared to placebo-treated hypoxic cells (45.33 ± 10.22% vs. 62.58 ± 1.41%, *p* < 0.001). In normoxic conditions, colchicine treatment showed a slight, non-significant reduction in IL-1β expression compared to untreated normoxic cells (44.06 ± 4.56% vs. 52.02 ± 7.89%, *p* > 0.05). These results suggest that colchicine’s inhibitory effect on IL-1β expression is more pronounced under hypoxic conditions, aligning with its potential therapeutic benefit in ischemic scenarios. The calculation of IL-1β expression is depicted in Figure 3.

## 4. Discussion

This study demonstrates that CoCl_2_ treatment significantly induces NLRP3 activation and IL-1β production in 3T3 cells. Targeting ASC-NLRP3 colocalization could offer a novel approach for anti-inflammatory therapy [24,25,26,27,28,29,30,31,32,33]. Apoptosis-associated speck-like protein containing a CARD (ASC) and NOD-like receptor protein 3 (NLRP3) are constituents of the inflammasome, a protein complex implicated in the inflammatory response and various diseases [34,35]. ASC functions as an adaptor in inflammasome assembly [36], while NLRP3 acts as a sensor detecting diverse stimuli such as oxidative stress, mitochondrial damage, or crystal accumulation, commonly observed in heart diseases like atherosclerosis, heart failure, or myocardial infarction [37]. Upon detection of these stimuli, NLRP3 interacts with ASC, leading to the activation of pro-caspase-1 and the subsequent formation of the inflammasome complex [38]. Activated caspase-1 catalyzes the processing of interleukin-1β and interleukin-18 precursors into their active forms [39] to stimulate the inflammatory response in heart diseases [40].

In this study, colchicine administration significantly reduced IL-1β expression, suggesting that colchicine can mitigate inflammasome activation under ischemic conditions by impeding ASC and NLRP3 interaction or availability [41]. Mechanistically, colchicine accomplishes this by interfering with speck formation, a crucial step in NLRP3 inflammasome activation, maybe through modulating the speck formation. Speck formation occurs when ASC oligomerizes and interacts with NLRP3 on the endoplasmic reticulum during inflammasome activation [42].

Colchicine further interferes with inflammasome assembly by disrupting the ASC–NLRP3 interaction, likely through its well-established effects on microtubule dynamics. Colchicine binds to tubulin, preventing microtubule formation, which is necessary for the trafficking of NLRP3 to interact with ASC [43,44]. This disruption could inhibit the movement and proximity of NLRP3 and ASC, essential for inflammasome assembly [45]. Additionally, by affecting speck formation, colchicine might inhibit ASC oligomerization, a critical step in the activation of the inflammasome [44].

Colchicine mitigates inflammation triggered by ischemia by indirectly inhibiting NLRP3 oligomerization, which reduces the production of IL-1β and IL-18. Unlike direct NLRP3 inhibitors like MCC950, colchicine’s effect on microtubule polymerization provides a unique mechanism of action [45]. This unique mechanism may contribute to colchicine’s broader anti-inflammatory effects and established clinical safety profile. This finding is consistent with studies showing colchicine’s ability to suppress NLRP3 inflammasome activation and decrease pro-inflammatory cytokines in cardiovascular contexts [46].

Colchicine’s binding to tubulin prevents microtubule formation, inhibiting NLRP3 inflammasome translocation and reducing active inflammasome numbers [47,48]. This disruption also interferes with the vesicular transport of pro-IL-1β, further decreasing IL-1β secretion [49]. Additionally, colchicine inhibits leukocyte recruitment and activation, crucial components of the inflammatory response [50]. Furthermore, colchicine modulates intracellular signaling pathways involved in inflammasome activation, including NF-κB, a key regulator of pro-inflammatory cytokine production and inflammasome gene expression [51]. It can also inhibit the TGF-β signalling pathway, either directly or by inhibiting TGF-β synthesis or secretion, resulting in decreased IL-1β production by myofibroblasts [52]. These multifaceted effects distinguish colchicine from more targeted NLRP3 inhibitors currently under development, potentially explaining its broader anti-inflammatory impact and well-established clinical safety profile.

The hypoxia-induced upregulation of TGF-β expression and activation stimulates fibroblast differentiation into myofibroblasts [25]. Excessive myofibroblast proliferation and differentiation may lead to undesirable scar tissue formation, disrupting normal organ or tissue function [53]. In the context of heart disease, myofibroblast proliferation and differentiation are associated with ventricular hypertrophy and myocardial fibrosis, contributing to decreased heart function and increased cardiovascular complications [26].

Our findings align with previous studies showing colchicine’s anti-inflammatory effects in cardiovascular diseases [54,55,56,57]. However, while we observed a significant reduction in IL-1β expression with colchicine treatment, the effect on ASC-NLRP3 colocalization, although reduced, did not reach statistical significance. This discrepancy warrants further investigation and may suggest additional mechanisms by which colchicine modulates inflammasome activity. Colchicine’s ability to disrupt ASC-NLRP3 colocalization and reduce IL-1β expression under hypoxic conditions provides a mechanistic explanation for its cardioprotective effects observed in clinical trials. For instance, the Colchicine Cardiovascular Outcomes Trial (COLCOT) demonstrated reduced cardiovascular events in patients with recent myocardial infarction treated with colchicine. Our results offer a cellular basis for these clinical observations, highlighting the potential of targeting inflammasome assembly as a therapeutic strategy in ischemic heart disease. By inhibiting inflammasome complex formation, colchicine decreases IL-1β production, potentially reducing inflammation and tissue damage associated with ischemia. Understanding the mechanisms of post-ischemic inflammation, particularly involving IL-1β and NLRP3, is crucial for developing effective therapies to mitigate tissue damage and improve prognosis in ischemic heart disease patients. This study lays the groundwork for further exploration of inflammasome colocalization in inflammation and the development of novel drugs targeting inflammasomes.

Future research should focus on elucidating colchicine’s effects on inflammatory cells concerning NLRP3 inflammasome; assessing potential benefits and risks in managing ischemic heart disease and developing therapies targeting IL-1β and NLRP3 pathways or inhibiting inflammasome activity. These efforts may yield innovative and more efficacious treatments for ischemic heart disease.

Previous research has demonstrated the effectiveness of colchicine in reducing inflammation in cardiovascular diseases, particularly by decreasing myocardial damage and improving outcomes in patients with acute coronary syndrome [53,54,55,56]. However, the novelty of our study lies in elucidating the cellular mechanism by which colchicine attenuates NLRP3 inflammasome activation in fibroblasts under hypoxic conditions. Unlike previous studies focusing on clinical outcomes, this study provides direct evidence of colchicine’s ability to disrupt ASC-NLRP3 colocalization at the cellular level, specifically in fibroblasts. This mechanism, targeting microtubule dynamics and inflammasome assembly, offers novel insights into colchicine’s broader anti-inflammatory effects beyond clinical settings.

## 5. Conclusions

Our study demonstrates that colchicine treatment significantly reduces IL-1β expression in hypoxia-induced 3T3 fibroblasts, likely by disrupting NLRP3 inflammasome activation. While the effect on ASC-NLRP3 colocalization was not statistically significant, the trend toward reduced colocalization suggests this as a potential mechanism of action. These findings provide new insights into the cellular mechanisms underlying colchicine’s anti-inflammatory effects in the context of ischemic heart disease. Future research should focus on validating these results in animal models of myocardial infarction and exploring the potential of targeting NLRP3 inflammasome assembly as a therapeutic strategy in cardiovascular diseases.

## Figures and Tables

**Figure 1 biomolecules-15-00367-f001:**
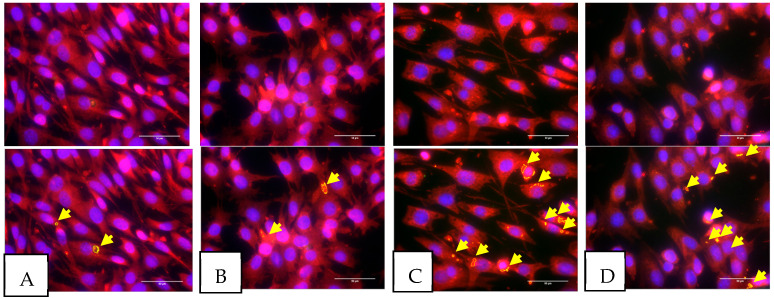
This figure illustrates immunofluorescence staining on 3T3 cells following various exposures, i.e., (**A**) normal condition treated with vehicle; (**B**) normal condition treated with colchicine; (**C**) ischemic condition treated with vehicle; and (**D**) ischemic condition treated with colchicine. ASC (labeled in red) and NLRP3 (labeled in green) protein co-localization configured yellow specks signifying ASC-dependent inflammasome activation. The bottom images are identical to the top images, with representative yellow arrows indicating ASC specks. All experiments were performed five times; figures are from one representative replicate. Images were taken at 40× magnification; scale bars are 50 µm. Created using BioRender.com (Science Suite Inc., Toronto, ON, Canada).

**Figure 2 biomolecules-15-00367-f002:**
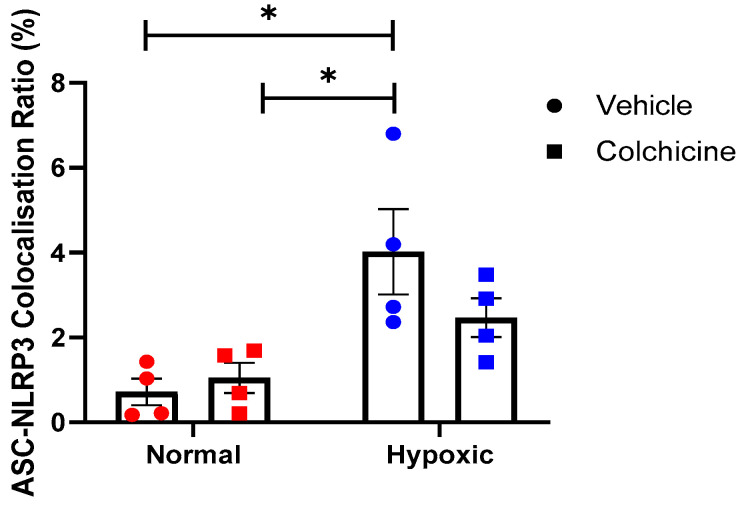
This figure illustrates colocalization speck intensity as a ratio of ASC fluorescence intensity, revealing ASC-NLRP3 complex formation. There is no significant difference in ASC-NLRP3 colocalization in colchicine vs. placebo within the same condition related to ischemic condition (cobalt chloride exposure). Red colors defined the normal condition, whereas blue colors defined the hypoxic/ischaemic condition. * The statistical significance was identified in the control group under ischemic conditions compared to normal conditions treated with either placebo-treated or colchicine-treated hypoxic cells (*n* = 5; *p* < 0.05).

**Figure 3 biomolecules-15-00367-f003:**
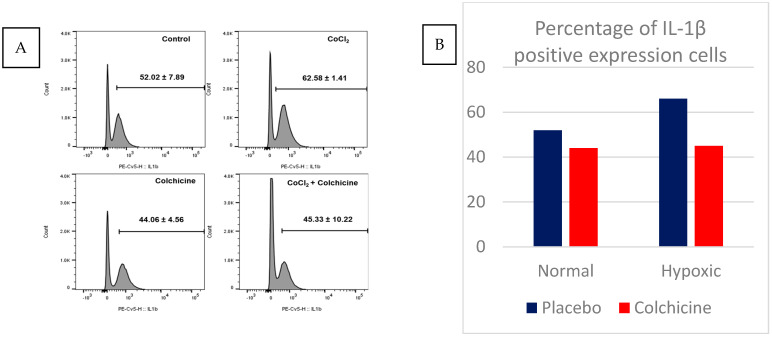
This figure illustrates the effect of colchicine treatment in normoxic and hypoxic 3T3 cells. (**A**) Absolute number of IL-1β-positive expressions among cell cultures from immunofluorescence staining; (**B**) relative number of IL-1β-positive expressions among cell cultures. The IL-1β expression was significantly reduced in colchicine-treated hypoxic cells compared to placebo-treated hypoxic cells (62.58 ± 1.41% vs. 45.33 ± 10.22%, *n* = 5; *p* < 0.001).

## Data Availability

The data presented in this study are available on request from the corresponding author.

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
