# Peer review of "The Emerging Role of Colchicine to Inhibit NOD-like Receptor Family, Pyrin Domain Containing 3 Inflammasome and Interleukin-1β Expression in In Vitro Models"

_biomolecules, 2025, doi:10.3390/biom15030367_

Round 1
Reviewer 1 Report
Comments and Suggestions for Authors
ASC serves as an adaptor protein that bridges NLRP3 with caspase-1 leading to the activation of caspase-1 and maturation of IL-1beta/IL-18. Authors tried to clarify the mechanism in which colchicine affect ASC-NLRP3 binding leading to less bridging with caspase-1 and reducing IL-1 beta expression in vitro cell model. Hypoxia treatment significantly induced ASC-NLRP3 colocalization (p<0.05), but the reduction of ASC-NLRP3 colocalization by colchicine treatment of hypoxic 3T3 cells was not statistically significant. However, IL-1β expression was significantly inhibited in colchicine-treated hypoxic 3T3 cells compared to those 41 treated with placebo (p<0.05).
- To strengthen the hypothesis of the study caspase-1 should be evaluated.
- In the Title in vivo model is right ?
- In Materials & Methods co-localization of ASC/NLRP3 was not described in detail.
- In Discussion too long.
- P6, line 260, incorrect description
Author Response
ASC serves as an adaptor protein that bridges NLRP3 with caspase-1 leading to the activation of caspase-1 and maturation of IL-1beta/IL-18. Authors tried to clarify the mechanism in which colchicine affect ASC-NLRP3 binding leading to less bridging with caspase-1 and reducing IL-1 beta expression in vitro cell model. Hypoxia treatment significantly induced ASC-NLRP3 colocalization (p<0.05), but the reduction of ASC-NLRP3 colocalization by colchicine treatment of hypoxic 3T3 cells was not statistically significant. However, IL-1β expression was significantly inhibited in colchicine-treated hypoxic 3T3 cells compared to those 41 treated with placebo (p<0.05).
Authors' response: thank you for your positive feedback. It is true that only IL-1β expression was significantly inhibited in colchicine-treated hypoxic 3T3 cells compared to those treated with placebo. The reduction of ASC-NLRP3 colocalization. While the effect on ASC-NLRP3 colocalization was not statistically significant, the trend towards reduced colocalization suggests this as a potential mechanism of action.
- To strengthen the hypothesis of the study caspase-1 should be evaluated.
Authors' response: thank you for your suggestion. It is well established that NLRP3 inflammasome activation triggers caspase-1 activation and IL-1β maturation. Unfortunately, our study doesn't evaluate the caspase-1 expression.
- In the Title in vivo model is right ?
Authors' response: thank you for your comment. We apologise for the errors. It should be in vitro.
- In Materials & Methods co-localization of ASC/NLRP3 was not described in detail.
Authors' response: thank you for your suggestion. We have added the description about co-localization of ASC/NLRP3 in the methods section.
- In Discussion too long.
Authors' response: thank you for your suggestion. We have shortened the discussion section by removing several sentences which are unnecessary.
- P6, line 260, incorrect description
Authors' response: thank you for your suggestion. We have deleted that incorrect description.
Reviewer 2 Report
Comments and Suggestions for Authors
The manuscript has the following issues:
-
In the caption of Figure 1, the magnification scale is missing.
-
In the captions of Figures 2 and 3, there is no description of "n=?" (sample size).
-
In Figure 2, it mentions "vehicle treatments," while in Figure 3, it refers to "placebo-treated hypoxic cells." Are these terms consistent?
-
The caption of Figure 3 does not explain what panels A and B represent.
-
The authors should investigate the dose-response and time-response relationships for the results shown in Figure 3.
The English could be improved to more clearly express the research.
Author Response
The manuscript has the following issues:
-
In the caption of Figure 1, the magnification scale is missing.Authors' response: the magnification scale can be seen in the bottom right corner of each image.
-
In the captions of Figures 2 and 3, there is no description of "n=?" (sample size).Authors' response: we added number of replication (n) in the description of Figure 2 and 3.
-
In Figure 2, it mentions "vehicle treatments," while in Figure 3, it refers to "placebo-treated hypoxic cells." Are these terms consistent?Authors' response: thank you for your feedback, we decided to replaced "vehicle treatments" with "placebo-treated hypoxic cells" to make it more consistent.
-
The caption of Figure 3 does not explain what panels A and B represent.Authors' response: thank you for your suggestion, we added the explanation of panel A and B.
-
The authors should investigate the dose-response and time-response relationships for the results shown in Figure 3.Authors' response: thank you for your suggestion, however it might be difficult to investigate the dose-response and time-response relationships at this momment
Round 2
Reviewer 1 Report
Comments and Suggestions for Authors
Well revised.
Author Response
Thank you very much for your valuable feedbacks and suggestions. I really appreciate your expertise and the time you took to make sure the manuscript improved better.
Reviewer 2 Report
Comments and Suggestions for Authors
The authors have completed the manuscript revisions. However, regarding the data points for Placebo and Colchicine in Figure 3b, they should be presented in distinct visual styles (e.g., different shapes or filled/unfilled markers) rather than identical ones to enhance clarity for readers.
Comments on the Quality of English LanguageThe English could be improved to more clearly express the research.
Author Response
Thank you for your suggestion. We have revised data points for Placebo and Colchicine in Figure 3b.